# Distinctive Features of Extracellular Vesicles Present in the Gastric Juice of Patients with Gastric Cancer and Healthy Subjects

**DOI:** 10.3390/ijms26125857

**Published:** 2025-06-18

**Authors:** Gleb Skryabin, Adel Enikeev, Anastasiia Beliaeva, Sergey Galetsky, Dmitry Bagrov, Andrey Moiseenko, Anna Vnukova, Oiatiddin Imaraliev, Ivan Karasev, Elena Tchevkina

**Affiliations:** 1Institute of Carcinogenesis, N.N. Blokhin National Medical Research Center of Oncology, Moscow 115522, Russia; g.skrjabin@ronc.ru (G.S.); a.enikeev@ronc.ru (A.E.); a.belyaeva@ronc.ru (A.B.); s.galetskiy@ronc.ru (S.G.); bagrov@mail.bio.msu.ru (D.B.); 2Faculty of Biology, Lomonosov Moscow State University, Moscow 119991, Russia; postmoiseenko@gmail.com (A.M.); biochem.fan@ya.ru (A.V.); 3Institute of Clinical Oncology, N.N. Blokhin National Medical Research Center of Oncology, Moscow 115522, Russia; o.imaraliev@ronc.ru (O.I.); i.karasev@ronc.ru (I.K.)

**Keywords:** gastric juice, small extracellular vesicles, exosomes, gastric cancer, cryo-electron microscopy, nanoparticle tracking analysis, tetraspanins, CD9

## Abstract

Extracellular vesicles (EVs) are key mediators of intercellular communication and play a vital role in cancer progression. While EVs in the blood are well-studied, those in local body fluids, such as gastric juice (GJ), remain underinvestigated. Previously, we first characterized GJ-derived EVs and demonstrated their potential for gastric cancer (GC) screening. Here, we conducted a detailed morphological analysis of GJ-EVs using cryo-electron microscopy, identifying both typical and atypical EV subtypes, and categorized their relative abundances. A subsequent comparison of the size distribution of GJ-derived EVs by nanoparticle tracking analysis revealed significant differences between samples obtained from GC patients (n = 40) and healthy subjects (n = 25). Additionally, the mean EV sizes differed significantly according to the presence of the tetraspanin protein CD9. Furthermore, the ratio of CD9-positive to CD9-negative EV samples differed between cancer patients and healthy donors. These data suggest that GJ contains distinct subpopulations of EVs that vary in size and CD9 expression, as well as EVs with certain types of atypical morphology. The identification of discrepancies in EV size and the presence of CD9 between GJ from cancer patients and healthy individuals offers potential avenues for the identification of new GC markers.

## 1. Introduction

Gastric cancer (GC) remains one of the most prevalent and deadly malignancies worldwide, with significant geographical variations in its incidence and mortality rates. According to the Global Cancer Observatory (GLOBOCAN) 2022 estimates, gastric cancer is the fifth most commonly diagnosed cancer and the fourth leading cause of cancer-related deaths globally [1]. In regions lacking early detection programs for gastric cancer, up to 80% of cases are diagnosed at advanced stages [2], with stage IV patients experiencing a dismal 5-year survival rate of less than 10% [3]. Conversely, in high-incidence areas such as Japan and South Korea, the establishment of systematic screening programs has markedly improved outcomes. In these regions, 50% of GC cases are detected at an early stage, leading to a remarkable 5-year survival rate of up to 90% and a significant reduction in GC-related mortality [4]. This stark contrast highlights the critical importance of early detection initiatives in improving patient prognosis and reducing the burden of gastric cancer.

Extracellular vesicles (EVs), in particular small EVs (sEVs) corresponding to exosomes, are secreted bilipid membrane-covered nanoscale particles known to be key players in intercellular communication involved in both physiological and pathological processes. The EV cargo carries molecular signatures that largely reflect their parent cells, providing insights into the state of cells and tissues. This makes them promising biomarkers for various diseases, including cancer. Given the increasing prevalence of cancer and the urgent need for early detection methods, research into EVs as diagnostic tools has gained momentum.

In the conventional approach to the screening of cancer markers, blood has been the most commonly utilized source of EVs. However, the utility of this method is constrained by the low proportion of cancer-derived exosomes present in the circulation, which can range from 0.2% [5] to 10% [6]. Furthermore, blood contains a considerable number of nanoscale non-vesicular particles, predominantly lipoproteins, which can contaminate sEV preparations [7,8]. These limitations have prompted the investigation of “local biofluids,” which are in direct contact with the tumor site and may provide a more concentrated source of tumor-derived EVs. This approach has been demonstrated to be effective, with the evidence provided by our research indicating that aspirates from the uterine cavity can serve as an effective source of EVs for the detection of ovarian cancer markers [9]. In the context of the search for gastric cancer markers, gastric juice appears to be a very promising source of EVs, as it is expected to be enriched in vesicles secreted by the primary tumor and the microenvironment and does not contain EVs originating from blood and immune system cells, which are the main EV producers in the bloodstream. Consequently, EVs isolated from gastric juice may exhibit a more specific molecular profile, offering more comprehensive insight into tumor-dependent molecular alterations.

Surprisingly, sEVs from gastric juice have been scarcely studied [10,11]. We were the first to isolate and characterize gastric juice sEVs in accordance with the ISEV (International Society for Extracellular Vesicles) guidelines, showing that gastric juice (GJ) is a promising source of extracellular vesicles, including those capable of carrying molecular markers for GC diagnosis [12]. However, since then, no new studies addressing any aspect of the biology of GJ-derived EVs have been published.

In our previous work, we investigated images of GJ EVs captured by transmission electron microscopy (TEM) and identified EVs with non-canonical morphologies, including elongated, multilayered and some other structures. In the present study, we employed cryo-EM to visualize GJ EVs, characterize their polymorphism, and assess the size distribution and the proportion of each morphological subtype.

In the context of searching for potential GC diagnostic markers, a comparison was performed of the size characteristics and protein composition of EVs isolated from the GJ of GC and non-cancer patients. Our findings reveal that EVs derived from the gastric juice of GC patients are significantly larger in size compared to those from healthy individuals. The findings from our previous pilot study suggest a relationship between the presence of CD9 tetraspanin, a widely recognized and utilized exosome marker, and the dimensions of EVs in GJ samples. The present study validated this observation using the expanded cohort under investigation. Moreover, an uneven distribution of CD9(–) EVs in GC patients and healthy individuals was demonstrated, with the control group exhibiting a significantly higher frequency of these samples.

## 2. Results

### 2.1. Characterization of GJ EVs by Cryo-EM

Previously, we demonstrated that sEVs could be successfully isolated from GJ using differential ultracentrifugation [12]. In this study, we have applied the same methodological approach with a larger sample size to conduct a comparative analysis of vesicle size in patients with GC and healthy individuals. This study comprised a total of 40 patients diagnosed with gastric adenocarcinoma and 25 healthy donors (HDs). The mean age of the patient group was 58.9 years (SD 11.7), while the mean age of the healthy control group was 48.2 years (SD 14.3).

The characterization of sEV samples was conducted in accordance with the guidelines of the International Society for Extracellular Vesicles (ISEV) [13], employing the following methods: transmission electron microscopy (TEM) for morphological analysis, nanoparticle tracking analysis (NTA) to evaluate concentration and size distribution, and western blot analysis for exosomal marker analysis.

TEM with negative staining showed a notable prevalence of particles with typical cup-shaped morphology (Figure 1A). Concurrently, we identified the presence of vesicles exhibiting atypical morphology, including elongated and multilayered vesicles (Figure 1B), consistent with our prior observations of GJ EVs [12].

Cryo-electron microscopy was used for the first time to assess the polymorphism of GJ-derived vesicles in more detail. Unlike ordinary TEM, this method allows the analysis of vesicles in their native state, unaltered by sample preparation. A sample of GJ EVs obtained from a healthy donor was examined; according to the TEM with negative staining, it contained some atypical particles (Figure 1B). A total of 267 cryo-EM images containing 968 extracellular vesicles (EVs) were analyzed manually by two independent observers to categorize and classify the EVs according to their shape, number of membrane layers, and the presence of inner secondary vesicles. Based on the morphological features, we were able to identify six different categories of EVs, of which 43% were represented by typical spherical vesicles surrounded by a single membrane, referred to as single vesicles. More than half (56%) of the EVs were represented by complex membrane structures, including double EVs, multi-layered EVs, dumbbell (or bowling pin)-shaped vesicles, and vesicles containing two or more daughter or ‘nested’ vesicles (Figure 2A, left). The latter category is referred to as vesicle ‘sacs’. Notably, the presence of EVs with similar atypical morphology was described previously for a number of body fluids. As ‘sacs’ accounted for almost 18% of all EVs, we decided to also examine the daughter vesicles contained within them (641 ‘nested’ vesicles in total) (Figure 2A, right). The distribution of morphotypes of ‘individual’ and ‘nested’ vesicles was found to be highly consistent; the only discrepancy observed was an increase in the proportion of multilayered EVs within the sacs at the expense of the >2-in-1 type. All EV types described are shown in Figure 2B. It should be noted that, since some EVs displayed features of multiple categories, the proportions of each morphological subtype should be considered approximate. Nevertheless, the discrepancy between the two sets of data obtained from the two observers was less than three percentage points.

In addition to the categorization of EVs, the mean size of EVs from each subtype was analyzed, both for ‘individual’ and ‘nested’ vesicles (Table 1).

Consistently, vesicle size increased with the number of additional layers, both for ‘individual EVs’ (Figure 2D, left) and ‘nested EVs’ (Figure 2D, right) (bowling pin-shaped EVs were excluded due to the small sample size). At the same time, the ‘2-in-1’ and ‘>2-in-1’ groups did not differ in size (Kruskal–Wallis test with Dunn’s post hoc test, *p* > 0.05). Interestingly, a notable pattern emerged when comparing the ‘individual’ and ‘nested’ vesicles of the same morphological type. The ‘individual’ EVs showed a significant increase in size compared to the ‘nested’ EVs of the same corresponding subcategory (Mann–Whitney *U*-test, *p* < 0.01). Histograms illustrating the size distribution of EVs within each category are shown in Figure 3A.

Similar atypical vesicle morphologies were observed in preparations from both healthy controls and gastric cancer patients. Although no formal group-based quantification was performed, the visual ratios of typical to atypical vesicles, and of the different morphological variants, were comparable in GC and control samples. Representative images of cancer samples can be found in Appendix A. Further comparative cryo-EM analysis using a larger sample size will reveal whether the morphological spectrum of GJ EVs obtained from GC patients differs from that of healthy donors.

Finally, we compared the size distribution data obtained by NTA and cryo-EM (Figure 3B). The results showed that the average particle size was larger in the latter case, due to a greater proportion of EVs measuring > 350 nm (19.8% vs. 6.5%). At the same time, a greater number of particles measuring between 40 and 70 nm was identified by cryo-EM, accounting for 18.9% of the total, compared to 7.9% detected by NTA. The fact that the median of the two distributions coincides seems to be due to this phenomenon.

### 2.2. Size of GJ EVs Differs Between CD9(+) and CD9(–) Samples

In addition to the analysis of EV morphology, the vesicular nature of the particles was further validated by enrichment of exosomal markers (Figure 4A). Following the ISEV guidelines, we examined the levels of several proteins from different intracellular compartments belonging to different functional categories. These included tetraspanin CD9, the ESCRT complex components TSG101 and Alix, and the lipid microdomain protein flotillin-2. As in our previous study [12], the presence of CD9 was detected in a subset of the samples. Notably, as we had previously observed, in those cases where it was detected, the presence of additional bands of reduced molecular weight was also observed. We also confirmed the presence of stomatin, a lipid raft protein belonging to the SPFH family (along with flotillin). We have previously proposed this protein as a highly specific marker for sEVs [14].

Finally, we compared the mean size distribution of EVs depending on the presence of CD9 using NTA (Figure 4B,C). According to our previous data from a pilot study of 13 samples, the mean particle size of GJ EVs differed depending on the presence or absence of CD9, as determined by western blotting [12]. This observation was further validated here in an expanded cohort (Figure 4B–D). The mean particle size for CD9(+) particles was determined to be 164.1 nm, while for CD9(−) particles, it was 136.4 nm (Mann–Whitney *U*-test, *p* = 0.002). Similarly, the average mode was found to be 114.6 nm for CD9(+) and 75.3 nm for CD9(−) particles.

### 2.3. EV Samples from GJ of GC Patients and Healthy Subjects Differ in Size and in the Presence of CD9

Next, we compared the NTA data on the size of GJ EVs from GC patients with those from healthy individuals and, surprisingly, found a significant difference (Figure 5A–C). Specifically, vesicles obtained from the GJ of cancer patients had a markedly larger mean diameter, with measurements averaging approximately 160.0 nm compared to 130.5 nm for vesicles from healthy controls (Mann–Whitney *U*-test, *p* = 0.037) (Figure 5C). Additionally, the mode of the vesicle size distribution also differed significantly between the two groups, with GC samples showing a mode of 104.7 nm, while healthy samples displayed a mode of 73.5 nm (Mann–Whitney *U*-test, *p* = 0.0014). This observation indicates differences in EV populations between GC and healthy donors, which may, in turn, be explained by the heterogeneity of EV types and the ratio of vesicles of different origins [15]. However, given the small sample size, this assumption requires further investigation using independent cohorts.

To test whether the distribution of CD9(+) and CD9(–) EVs differed between the healthy and cancer groups, the chi-squared test was applied. The results showed a statistically significant discrepancy in the distribution of EVs (*p* = 0.0082), with the population of normal samples consisting predominantly of CD9(–) EVs. In contrast, the distribution of CD9(+) and CD9(–) EVs in the GC group exhibited a comparable frequency (Figure 5D). Additionally, we examined the correlation between the presence of CD9 in GJ EVs and disease stage using Fisher’s exact test and the Cochran–Armitage trend test. No significant difference in CD9 presence was found between stages (*p* = 0.67), nor was there a significant trend (*p* = 0.31) in the current data. However, due to the small sample size, particularly for stages I and II, these results should be interpreted with caution (Appendix A). We also tested whether vesicle size depended on the stage of the disease. One-factor analysis of variance (ANOVA) revealed no significant difference in the mean EV size across the different stages (*p* = 0.27) (Appendix A).

A two-factor ANOVA was then used to determine the significance of the contribution of each factor, i.e., clinical status (diagnosed GC) and presence of CD9, to the observed differences in EV size. This analysis revealed that both clinical (*p* = 0.0058) and CD9 status (*p* = 0.0003), as well as their interaction (*p* = 0.0251), had a statistically significant effect on the size of GJ EVs. The influence of CD9 status was found to be the most significant (18% of the variation), followed by clinical status (10%) and the interaction of factors (6%). At the same time, Tukey-corrected multiple comparisons indicated that the HD CD9(–) sample group showed a significant deviation from all other groups (Figure 5E). The most pronounced discrepancies between groups were observed between GC CD9(+) and HD CD9(–) (*p* < 0.0001) and between HD CD9(+) and HD CD9(–) (*p* = 0.0006). Although the factorial interactions contribute less to the variation (6%), the main effects of each factor (health status and CD9 status) had a more significant impact on GJ EV size.

## 3. Discussion

Despite the fact that the presence of extracellular vesicles, including small exosome-like vesicles, has been detected in almost all body fluids, research conducted on gastric juice-derived EVs has been limited in scope. In a previous pilot study, we first demonstrated a high degree of heterogeneity of GJ-derived EVs in terms of morphology and the composition of exosomal protein markers.

Cryo-EM has emerged as a powerful tool in the field of EV research, gaining significant traction due to its unique capabilities compared to conventional imaging methods. Cryo-EM preserves the native state of EVs, thereby avoiding artifacts and eliminating the necessity for staining [16,17]. The lipid bilayers have relatively high contrast in the cryo-EM images, so the EVs can be easily distinguished from possible admixtures and contaminants, which are frequently present in samples. This characteristic is of particular benefit when studying surface features and identifying subtle differences among EV subpopulations. The versatility and accuracy of cryo-EM have led to its widespread adoption as a method for investigating the structural and morphological diversity of EVs in various biological contexts.

To date, cryo-EM has been employed to characterize EVs isolated from a wide range of body fluids including plasma/total blood, cerebrospinal fluid, ejaculates, and follicular fluid, as comprehensively reviewed by Saadeldin et al. [18]. Previously, our research group has contributed to this growing body of work by being the first to apply cryo-EM to EVs isolated from uterine aspirates [9]. However, the morphology of GJ-derived EVs has not been studied using this method yet. Here, we used a pilot sample to assess the polymorphism of GJ EVs and classify them into distinct and recurrent subcategories. Specifically, we identified the presence of unilamellar (single, oval-, and bowling pin-shaped vesicles), multilamellar (double- and multimembrane vesicles), and even multicompartmental structures—vesicles containing other vesicles (sacs). Importantly, the morphological subcategories identified among GJ-derived EVs closely resemble those described by other researchers for vesicles of different origins. In the literature, there still has been minimal inquiry into the differences between unilamellar and multilamellar EVs in terms of their biogenesis mechanisms and functional effects. Currently, we can only speculate about the causes and functions of multilamellar EVs.

Our analysis of 968 individual EVs revealed a remarkable heterogeneity, with six distinct morphological categories identified. It is noteworthy that the proportion of multilamellar vesicles in GJ EVs is strikingly high compared to other biofluids. In most studies of EVs from fluids such as plasma [19], cerebrospinal fluid [20], or ejaculates [21], single vesicles typically dominate, often accounting for 70–80% of the total EV population [18]. In contrast, our analysis revealed that single vesicles constituted only 43% of GJ EVs, while multilamellar and other complex structures made up the majority (56%). This high prevalence of multilayered vesicles in gastric juice may reflect unique biophysical or biochemical conditions of the stomach, such as the acidic pH or the presence of digestive enzymes, which could influence EV biogenesis and morphology, suggesting a specialized role for multilamellar EVs in the gastric environment, potentially related to cargo protection, or targeted delivery within the harsh conditions of the stomach.

In light of the aggressive environment of gastric juice, it can be posited that a high abundance of multilamellar vesicles is designed to ensure enhanced cargo protection. Multilamellar EVs, similar to synthetic multilayered liposomes, may enable the controlled and sustained release of cargo through layer-by-layer degradation or stimulus-sensitive triggers [22,23,24] while also allowing the increased loading of hydrophobic molecules within their bilayers. These structural features suggest that multilamellar EVs could prolong intercellular signaling and enhance the delivery of bioactive components. The presence of multiple layers may also slow cargo release, extending the duration of intercellular communication. Additionally, multilamellar EVs might supply recipient cells with lipids to repair pathogen-induced membrane damage.

Pathological conditions, such as Gaucher disease, prion infection, and bacterial lipopolysaccharide exposure, have been shown to alter the formation and abundance of multilamellar EVs, indicating a potential link between disease states and EV morphology [25,26,27]. Overall, these findings suggest that EV subtypes may perform specialized functions in health and disease. However, further research is needed to fully understand their biological significance.

A particular emphasis should be placed on multilayered extracellular vesicles. Despite the absence of a consensus on the origins of these morphological forms, it can be stated with a high degree of confidence that at least some of the observed structures are present in their native form and are not artifacts of the isolation or storage methods. The presence of multilayered EVs has been detected in both fresh samples and samples isolated using various techniques, including ultracentrifugation and tangential flow filtration [28]. While the formation of multilayered EVs may be influenced by certain isolation techniques, their presence in unprocessed biofluids and samples stored in cryoprotective buffers suggests a natural origin. Several hypotheses have been proposed to explain their biogenesis, including the encapsulation of smaller vesicles within larger ones or simultaneous membrane budding (a topic more thoroughly delineated in Broad’s paper [28]). However, further research is necessary to fully understand the mechanisms of their biogenesis, functional significance, and potential applications in diagnostics and therapy.

A particularly intriguing observation in our study is the high prevalence of vesicle sacs containing nested vesicles. To the best of our knowledge, no previous studies have characterized the morphology of the vesicles contained within other vesicles. Remarkably, the morphological distribution of these nested vesicles closely mirrored that of individual EVs, with the exception of a slight increase in multilayered particles within the sacs. The size distribution analysis revealed that EVs increased in size with the number of additional layers, both for individual and nested vesicles. This finding aligns with previous studies demonstrating that multilayered EVs tend to be larger than single-layered ones [20,29]. Interestingly, individual EVs were significantly larger than their nested counterparts within the same morphological category, suggesting that the encapsulation process may influence vesicle size. It is difficult to say unambiguously what these differences are related to. However, it is reasonable to assume that the molecular cargo compositions of nested and single vesicles differ. This is also evidenced by the differences in electron density between some nested and single vesicles, observed in our and other studies [21,30,31]. Although variation in size and molecular composition has been repeatedly demonstrated even for EVs of single origin, the exact relationship between these characteristics remains unclear. Depending on their size, EVs may be enriched with certain molecules, such as circulating tumor DNA, as has been demonstrated for vesicles in the plasma of prostate cancer patients [32]. It can also be assumed that the differences in size between single and nested vesicles are related to the mechanism of their acceptance by recipient cells. For example, it has been shown that EVs with a larger volume and greater numbers of molecules do not penetrate biological barriers or enter target cells as effectively as smaller EVs [33]. Since small EVs are most likely to enter recipient cells via endocytosis and larger EVs via fusion with the plasma membrane, it is conceivable that smaller vesicles are packaged into multivesicles or vesicular sacs for more efficient and/or simultaneous delivery to recipient cells. Additionally, it is plausible that physical processes are responsible for this phenomenon: extracellular vesicles exhibit a higher proportion of lipid rafts in comparison to cell plasma membranes. These rafts are enriched in specific lipids, specifically, phosphatidylcholine, which facilitates membrane fusion at lower pH levels [34]. pH acidification was shown to trigger both viral [35,36] and EV fusion [37]. It is also noteworthy that while the differences in size distributions between cryo-EM and nanoparticle tracking analysis (NTA) were not drastic, the mean size of EVs did show some variation between the two methods. Cryo-EM detected a greater proportion of larger EVs (>350 nm) and smaller particles (40–70 nm) compared to NTA, which likely reflects the limitations of NTA in accurately resolving these size ranges. Larger vesicles are often excluded as noise in NTA, while smaller particles may be underrepresented due to the technique’s sensitivity threshold. Importantly, although the average vesicle size was larger in the cryo-EM analysis than in the NTA, the mode value was higher according to the NTA data. This suggests that the increase in mean vesicle size observed in the cryo-EM analysis was primarily due to the presence of large vesicles exceeding 400 nm, which appeared to be beyond the sensitivity threshold of the NTA method. Previous studies have reported differences in vesicle size when assessed by different methods. For instance, significant discrepancies in the size distribution of EVs released by human breast cancer MCF7 cells were observed when various high-resolution imaging techniques were employed. In particular, the size distribution estimated by NTA shows fewer particles in the under 100 nm and over 1000 nm populations than scanning electron microscopy (SEM). The proportion of EVs over 500 nm in diameter from the total population was 1.2% in NTA, 2.7% in EM-tomography, and 17.9% in SEM analysis [38].

The NTA method was routinely employed to obtain data on the size distribution and concentration of particles in all obtained samples of GJ EVs. When these data were compared between the GC and HD groups, it was found that the samples of EVs isolated from the GJ of gastric cancer patients were significantly larger than those of healthy donors. This discrepancy may be linked to differences in the proportion of EV subpopulations. Thus, previously, in our pilot study, we showed that vesicles expressing tetraspanin CD9 (one of the most common exosomal markers), as detected by western blotting, were larger than EVs in which CD9 was not detected [12]. In the current study, using an expanded cohort of 65 samples, we confirmed that CD9(+) EVs are significantly larger in mean size, mode, and median compared to CD9(−) EVs.

These findings highlight the importance of tetraspanins in shaping the physical and functional properties of EVs, with potential implications for their role in intercellular communication and disease progression.

Tetraspanins (TSPANs) are a family of transmembrane proteins found in nearly all cell types. They are characterized by their four transmembrane domains, connected by two extracellular loops and an intracellular loop, with short N-terminal and C-terminal cytoplasmic tails, which vary between different TSPANs. Members of this family, such as CD9, CD63, CD81, and CD82, form specialized complexes called tetraspanin-enriched microdomains (TEMs). These TEMs act as platforms for organizing proteins and lipids, facilitating interactions that are crucial for various cellular processes [39].

TSPANs have been found to be involved in a wide range of cellular processes, including cell adhesion, migration, signaling, membrane fusion, viral and bacterial infections, and in cancer (described in greater detail in reviews [39,40]). By organizing TEMs, they facilitate the assembly of multi-molecular complexes that regulate interactions between membrane proteins, lipids, and signaling molecules. This organization is critical for processes such as immune cell activation, fertilization, and tissue repair. Furthermore, TSPANs have been demonstrated to modulate membrane dynamics, influencing the rigidity, curvature, and stability of cellular membranes. Their capacity to interact with other TSPANs and partner proteins, such as integrins and growth factor receptors, further underscores their importance in coordinating cellular responses to external stimuli.

TSPANs have been demonstrated to play a critical role in the regulation of fusion-dependent processes. These processes range from myoblast formation and the formation of viral-induced syncytia [41] to sperm-egg fusion: CD9 has been shown to be almost essential for mammalian fertilization [42]. This involvement in membrane fusion extends to somatic cells, where experiments have shown that treatment with antibodies against TSPANs such as CD9, CD81, and CD151 reduces EV uptake by target cells [43]. This suggests that TSPANs play a vital role in the regulation of adhesion and fusion events.

In the context of EVs, TSPANs are often used as markers for exosomes, but their role extends far beyond mere identification. They have been demonstrated to play a pivotal role in diverse aspects of EV biology, including EV biogenesis, cargo selectivity, and cell targeting and fusion [44]. It has been established that they contribute to the membrane curving process, which results in the formation of intraluminal vesicles within multivesicular bodies. This, in turn, leads to the subsequent release of EVs into the extracellular space.

Thus, the observed difference in size between CD9(+) and CD9(–) EV samples may reflect distinct subpopulations of EVs that differ in terms of their biogenesis and functional activity. For instance, the increased size CD9(+) EVs could partially originate from vesicle fusion, given the aforementioned role of CD9 in membrane fusion events. The process of vesicle fusion has recently been visualized using cryo-EM, demonstrating that this fusion occurs in a pH- and protein-dependent manner [37]. CD9 could be involved in membrane fusion through several potential mechanisms involving its role in membrane organization, protein scaffolding, and membrane dynamics modulation. It may organize the plasma membrane by sensing or inducing curvature, thereby facilitating the close apposition of membranes required for fusion. For instance, it accumulates in regions of sperm–oocyte adhesion, where inverted conical CD9 structures directly affect the membrane curvature [45]. Thus, CD9 might be required to stabilize and/or further generate a more extreme membrane curvature that is required for the fusion step. Additionally, CD9 can act as a scaffold protein, linking critical components necessary for fusion events, such as viral receptor DPP4 and protease TMPRSS2 in the case of MERS-CoV infection [46].

At the same time, the difference in size between CD9(+) and CD9(–) EVs is more likely to be attributed to the original type of biogenesis. The two best-known types of EV biogenesis are the direct budding of EVs from the plasma membrane of parental cells, resulting in microvesicle (MV, or ectosome) secretion, and the formation of intraluminal vesicles within the endosomal trafficking system. This pathway of biogenesis leads to the secretion of exosomes. While MVs were previously thought to be significantly larger than exosomes, it has now been demonstrated that their sizes largely overlap. In the context of our results, it is important to note the recently demonstrated involvement of CD9 in forming small ectosomes that bud from plasma membrane vesicles. The average size of these vesicles is comparable to that of exosomes [47]. The authors provide evidence that small EVs bearing tetraspanins, especially CD9 and CD81 with little CD63, bud mainly from the plasma membrane, whereas others bearing predominantly CD63 with little CD9 form in internal compartments and qualify as exosomes [48]. Therefore, CD9(+) EVs may represent small ectosomes that bud from the plasma membrane and may be larger than exosomes. In contrast, CD9(–) vesicles may correspond to exosomes in a biogenesis-like manner.

Apart from that, the tumor microenvironment in GC patients, characterized by constant inflammation, may influence vesicle structure, potentially resulting in larger tumor-derived EVs. Thus, given that intratumoral hypoxia stimulates the secretion of EVs, especially MVs [49], it is possible that this could result in increased EV size and the accumulation of CD9-positive microvesicles (small ectosomes). In addition, GC patients may have elevated levels of *Helicobacter pylori*, which is also capable of releasing “outer membrane vesicles”, 20 to 300 nm in diameter, introducing additional variables into the results [50].

However, it should be noted that the observed differences in the size of CD9(+) and CD9(–) EV samples are related to the average size of the vesicle population within each sample. Further studies are needed to understand how the size of individual EVs differs depending on the presence of CD9. For example, immunolabeling of EVs followed by electron microscopy would be appropriate. Immunoprecipitation of CD9(+) EVs could also facilitate comparison of the CD9(+) and CD9(–) subpopulations. This would confirm the association of CD9 with EV size and reveal its relationship with the morphological variants of the vesicles. It could also help to determine whether differences in EV size are associated with morphology or CD9 expression independently.

In addition to differences in vesicle size, we also observed an uneven distribution of CD9-positive and CD9-negative EVs between groups. Notably, the proportion of CD9-positive samples differed between healthy donors and gastric cancer patients. As the composition of adhesion molecules (including tetraspanins) on EV membranes largely determines intercellular communication and contributes to an aggressive tumor phenotype, the presence of CD9-positive EVs could plausibly promote tumor progression.

In addition to being involved in cell adhesion and trafficking and vesicular and cellular fusion, CD9 is implicated in a variety of both intracellular and intercellular interaction processes. Alterations in the expression, localization, and vesicular secretion of CD9 have been identified in a variety of neoplasms. Although it was initially assumed to have rather tumor-suppressing properties, there is now a growing body of evidence of its association with the increased proliferation, migration, and survival of several cancer histotypes [51].

Overexpression of CD9 was found in ovarian cancer compared to benign tissues [52] and was found to promote the development of breast cancer bone metastasis [53]. Moreover, CD9 expression may have prognostic significance for lobular carcinoma [54] and invasive breast carcinoma, depending on the histology of the CD9-expressing cells (tumor or stromal immune cells) and the molecular subtype [55]. In the case of gastric carcinoma, increased CD9 expression has been observed in primary tumors and metastatic lesions, and a correlation has been demonstrated with vessel invasion, lymph node metastasis, and advanced stage [56]. Other studies performed on large cohorts of GC patients show that CD9-positive cases are significantly correlated with scirrhous-type GC, lymph node metastasis, and venous invasion. The five-year survival rate of patients with CD9-positive tumors was significantly lower than that of patients with CD9-negative tumors. In line with our findings, it is of particular interest that CD9-positive EVs from cancer-associated fibroblasts stimulate the migration and invasion of scirrhous-type gastric cancer cells [57]. Given the role of CD9 expression in GC progression, the increased presence of CD9-positive vesicles in the gastric juice of cancer patients may have significant diagnostic and prognostic implications. This highlights the importance of further research into the molecular composition of GJ-EVs and the potential of EV-encapsulated CD9 for GC diagnosis.

## 4. Materials and Methods

### 4.1. Clinical Specimens and Sample Processing

The collection of clinical material was conducted from 1 February to 20 December 2024, during a standard gastroscopic examination in the Endoscopy Department of the N.N. Blokhin National Medical Research Center for Oncology in accordance with the previously described protocol [12]. The study protocol was approved by the Ethics Committee of the N.N. Blokhin National Medical Research Center of Oncology (Ethics Committee Permission Protocol No. 1 for Project No. 24-25-00052 dated 25 January 2024). All experiments were conducted in accordance with the principles set forth in the Declaration of Helsinki. Prior to the commencement of the study, written informed consent was obtained from all participants. The initial volume of the GJ samples ranged from 2.5 to 10 mL. The samples were then diluted in 5 mL of ice-cold PBS immediately after collection. Within one hour of collection, the sample was cleared by centrifugation at 2000× *g* for 30 min at 4 °C. The supernatant was then diluted to 35 mL with ice-cold PBS and further cleared by centrifugation at 12,000× *g* for 45 min at 4 °C. The resulting supernatant was frozen at −80 °C until EV isolation.

### 4.2. EV Isolation

The small EVs were isolated using differential ultracentrifugation in accordance with the protocol that had been previously established [12]. Briefly, thawed samples were centrifuged at 12,000× *g* for 30 min at 4 °C, followed by the sedimentation of EVs at 100,000× *g* for 2 h at 4 °C. The resulting pellet was diluted in 5 mL of ice-cold PBS and re-sedimented at 100,000× *g* for 90 min at 4 °C. The final cleared pellet was resuspended in 120 μL of ice-cold PBS and aliquoted in Protein LoBind tubes (Cat No. 0030108434, Eppendorf AG, Hamburg, Germany) for NTA, TEM, and protein analysis. Aliquots were frozen in liquid nitrogen and stored at −80 °C for further analysis.

### 4.3. EV Visualization

The morphology of the EVs was analyzed using transmission electron microscopy (TEM). For this purpose, the carbon-coated TEM grids (Ted Pella, Redding, CA, USA) were subjected to a 45 s treatment using an Emitech K100X glow discharge system (Quorum Technologies Ltd., Laughton, UK) to render the carbon surface hydrophilic and enhance vesicle adsorption. The samples were applied to the grids for 30–60 s, stained twice with 1% uranyl acetate for 45 s each, and then dried. EVs were imaged using a transmission electron microscope JEM-1400 (JEOL Ltd., Akishima, Japan) at 120 kV (at least 10 fields of view per sample).

For cryo-electron microscopy (cryo-EM), a 3 µL aliquot of the sample was placed onto a lacey carbon grid with a perforated support film (EMCN, Beijing, China). Prior to use, the grid was treated with glow discharge using an EasyGlow system (TedPella, Redding, CA, USA). The grids were then flash-frozen in liquid ethane with an EM GP2 plunger (Leica Microsystems, Wetzlar, Germany), applying a blotting duration of 12 s. The imaging was conducted at an electron dose of 40 e/Å^2^ and a defocus setting of −2 μm, utilizing a JEM-2100 TEM (JEOL, Tokyo, Japan) fitted with a DE-20 camera (Direct Electron, San Diego, CA, USA). The entire imaging process was automated through SerialEM software (version 3.8). The resulting images were analyzed and processed manually using ImageJ (version 1.54f). Given the absence of a standardized classification for EV morphological subtypes observed by cryo-EM, we based our categorization on prior high-resolution studies [21,29] describing common structural features such as multilamellar vesicles and vesicle sacs. Manual classification was performed independently by two observers.

### 4.4. Nanoparticle Tracking Analysis

The quantity and size distribution of extracellular vesicles were measured in all samples using the NanoSight LM10 HS instrument (Malvern Panalytical Ltd., Malvern, UK) equipped with a NanoSight LM14 unit with onboard temperature control (Malvern Panalytical Ltd.), an LM 14C (405 nm, 65 mW) laser unit, and a high-sensitivity camera with a scientific CMOS sensor (C11440-50B, Hamamatsu Photonics, Hamamatsu City, Japan), following the previously established protocol [58]. In summary, the sample was exposed to the imaging process for a duration of one minute on 12 occasions. The obtained videos were processed using NTA software, version 2.3 build 33. This was followed by the summation of the results to generate a particle size distribution histogram, along with the concentration and key particle size data.

### 4.5. Immunoblotting and Antibodies

To determine the concentration of total protein in EV samples and cells lysed in RIPA buffer, the NanoOrange™ protein quantification kit (Cat. No. N6666, Lot No. 2149282, ThermoFisher Scientific, Eugene, OR, USA) was used according to the manufacturer’s recommendations using a SpectraMax M5e microplate reader (Molecular Devices, LLC, San Jose, CA, USA). Immunoblotting was performed as previously described [8] with minor modifications: in our study, we used 3 μg of total protein, which was applied to SDS-PAGE, and the proteins were visualized using SuperSignal™ West Femto Maximum Sensitivity Substrate (#34095, ThermoFisher Scientific, Rockford, IL, USA). The following primary and secondary antibodies and dilutions were used: anti-Flotillin-2 (#3436S, 1:1000; Cell Signaling Technology, Topsfield, MA, USA), anti-CD9 (#13174, 1:2000; Cell Signaling Technology), anti-TSG-101 (ab125011, 1:5000; Abcam, Cambridge, UK), anti-PCNA (#sc-7907, 1:500; Santa Cruz Biotechnology, Dallas, TX, USA), anti-Stomatin (#sc-134554, 1:500; Santa Cruz Biotechnology), anti-mouse goat polyclonal antibodies (#2367, 1:5000; Cell Signaling Technology); and anti-rabbit goat polyclonal antibodies (#29902, 1:80,000; Cell Signaling Technology).

### 4.6. Statistics

Based on the NTA-measured particle size and concentration, values of the mean, mode, percentile data (10th and 90th), standard deviation, and confidence interval were calculated using Wolfram Mathematica software ver. 11 (Wolfram Research, Champaign, IL, USA). The Kruskal–Wallis test with Dunn’s post hoc test was used to compare the size data of different morphological EV classes obtained using cryo-EM images. The non-parametric Mann–Whitney *U*-test was employed to make a comparison between the groups of “individual” and “nested” EVs and the “GC” and “HD” EVs. The chi-squared test was employed to assess the distribution of CD9 expression among the patient and healthy donor groups. For comparisons of NTA data among multiple groups, two-way analysis of variance was employed, followed by Tukey’s post hoc test.

## 5. Conclusions

Here, we present the first evidence of the high morphological heterogeneity of EVs in gastric juice. By analyzing the native structure of the vesicles using cryo-EM, we identified several atypical EV morphology variants that resemble those previously observed in vesicles from other biological sources.

Summarizing the results of the study of EV size and CD9 occurrence in the samples from GC patients and healthy subjects, the following conclusions can be drawn: (1) EVs from gastric cancer patients were significantly larger compared to those from healthy controls; (2) CD9(+) EVs were consistently larger than CD9(−) EVs; and (3) the distribution of CD9(+) and CD9(−) EVs differed markedly between the two groups, with control samples predominantly consisting of CD9(−) EVs, while GC samples exhibited a more balanced distribution. In other words, the size of GJ-derived EVs is significantly related to both the CD9 status and clinical status. Taken together, these observations suggest the presence of different EV subpopulations in gastric juice, with different ratios between gastric cancer patients and healthy individuals. The observed differences in EV size and CD9 representation in gastric juice-derived vesicles can contribute to the development of new liquid diagnostic markers for gastric cancer. We also demonstrate the high potential of EVs from local, tissue-specific bodily fluids for cancer diagnosis.

## Figures and Tables

**Figure 1 ijms-26-05857-f001:**
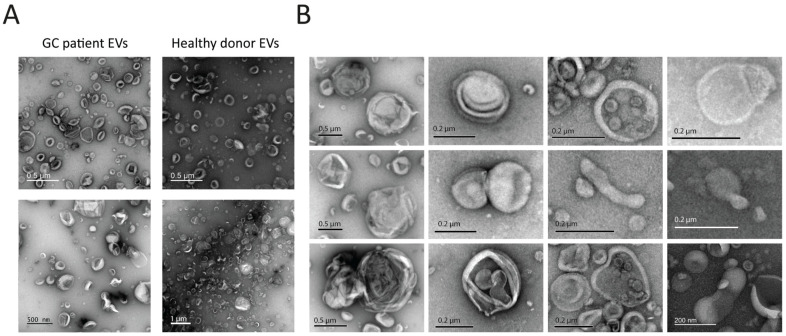
Visualization of GJ EVs by TEM. (**A**) TEM images depicting particles with typical EV size and morphology from GC patients and healthy donors. (**B**) Examples of vesicles with atypical morphology visualized by TEM.

**Figure 2 ijms-26-05857-f002:**
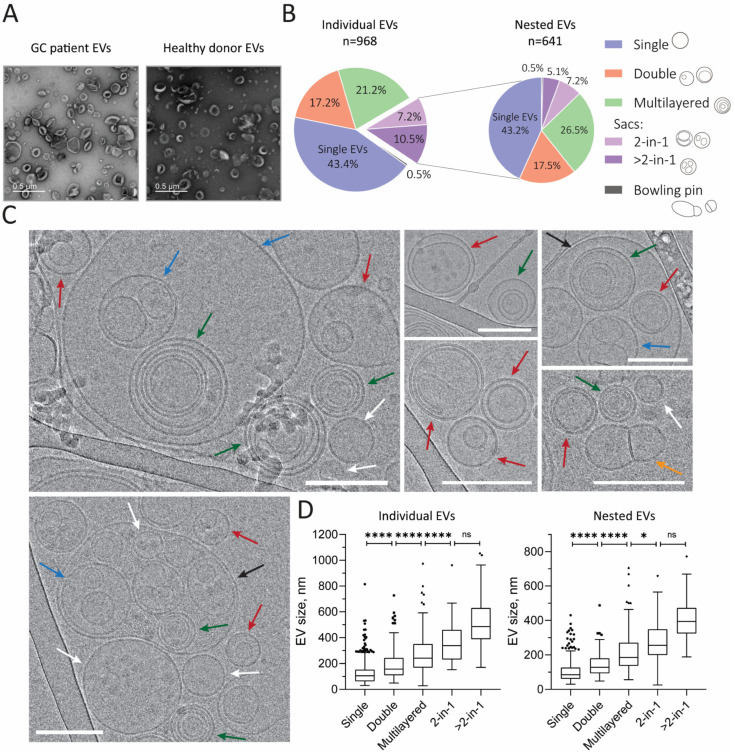
(**A**) Diversity of morphological types of GJ EVs. (**B**) The prevalence of each EV morphological subcategory, n = sample size. (**C**) Cryo-EM images illustrating the morphological diversity of GJ EVs. Micrographs of single (white arrows), double (red), multilayered (green), “2-in-1” (blue), “>2-in-1” (black), and dumbbell-shaped (orange) vesicles are shown. Scale bars 200 nm. (**D**) Sizes of each individual (left) and nested (right) EV morphological subcategory (**** *p* < 0.0001, * *p* < 0.05, ns—not significant). Graphs are plotted using the Tukey method.

**Figure 3 ijms-26-05857-f003:**
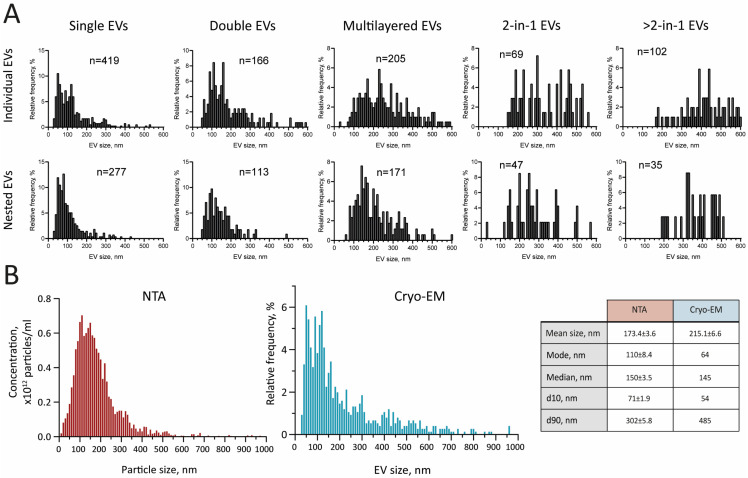
EV size distribution. (**A**) The size distribution histograms of the distinct EV morphology subtypes of ‘individual’ EVs (top row) and ‘nested’ EVs (bottom row). Data were obtained from cryo-EM images. n = sample size. (**B**) Comparison of size distribution data obtained from NTA (left) and cryo-EM (right). For distribution based on cryo-EM, only ‘individual’ EVs were included in the graph. Symbol ± indicates SEM.

**Figure 4 ijms-26-05857-f004:**
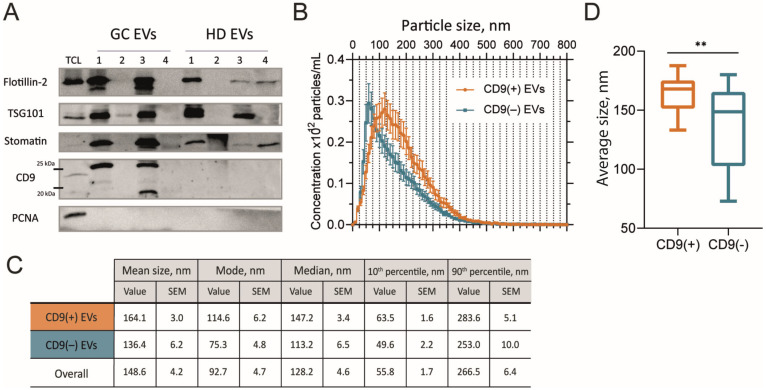
Characterization of EVs isolated from gastric juice. (**A**) Western blot analysis of exosomal markers flotillin-2, TSG101, stomatin, and CD9 in GC and HD EVs. Tumor cell lysate (TCL) obtained from gastric adenocarcinoma SNU-1 cell line was used as a control. The PCNA protein was used to confirm the absence of cellular proteins of non-vesicular origin in EV preparations. (**B**) NTA data on mean EV size distribution of a population of CD9(+) and CD9(–) samples. Error bars indicate SEM. (**C**) Mean values for EV size characteristics of the CD9(+) and CD9(–) samples and entire data according to NTA data. (**D**) Comparison of the mean size of CD9(+) and CD9(–) EVs (** *p* < 0.01).

**Figure 5 ijms-26-05857-f005:**
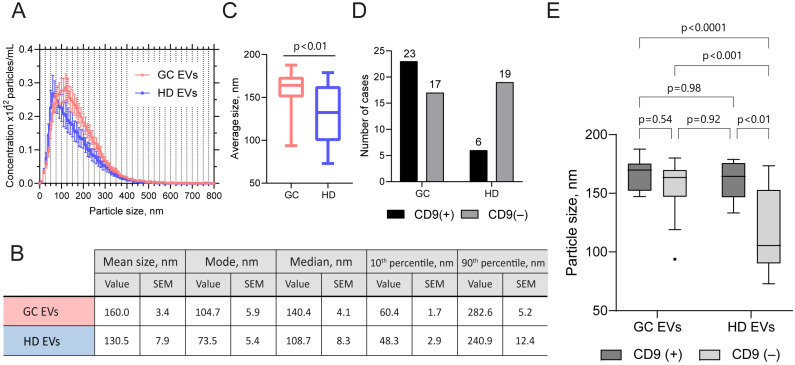
The characterization of GJ-derived EVs by nanoparticle tracking analysis. (**A**) NTA data on the mean EV size distribution of a population of gastric cancer (GC) and healthy donor (HD) samples. Error bars indicate SEM. (**B**) Mean values for EV size characteristics over the GC and HD samples according to NTA data. (**C**) Comparison of the mean size of GC and HD EVs. Bars indicate min to max. (**D**) Distribution of CD9(+) and CD9(–) EVs over the GC and HD groups. (**E**) Comparison of the mean size of CD9(+) and CD9(–) EVs in the GC and HD groups. The graph is plotted using the Tukey method.

**Table 1 ijms-26-05857-t001:** Mean values for EV size characteristics obtained using cryo-EM images. Symbol ± indicates SD.

	Individual EVs					Nested EVs				
	Single	Double	Multilayered	2-in-1	>2-in-1	Single	Double	Multilayered	2-in-1	>2-in-1
Mean size, nm	127.8 ± 94.5	191.3 ± 121.5	276.3 ± 147.7	361.3 ± 151.3	513.5 ± 190.2	107.1 ± 69.1	143.4 ± 70.39	215.8 ± 113.1	282.3 ± 104.4	405.3 ± 132.5
Mode, nm	87	164	171	419	430	61	89	112	N/A	324
Median, nm	105	158	241	338	485	84	128	185	255	394
d10, nm	46	84	123	185	297	47	71	105	144	216
d90, nm	257	356	478	530	786	202	233	359	494	627

N/A—the sample presents two values equally, so no value is given.

## Data Availability

The original contributions presented in this study are included in the article/Appendix A. Further inquiries can be directed to the corresponding author.

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
