# Peer review of "Distinctive Features of Extracellular Vesicles Present in the Gastric Juice of Patients with Gastric Cancer and Healthy Subjects"

_ijms, 2025, doi:10.3390/ijms26125857_

Round 1

Reviewer 1 Report

Comments and Suggestions for Authors

This study utilized a cryo-electron microscopy system to analyze the morphological heterogeneity of extracellular vesicles (EVs) in gastric fluid, revealing significant differences in size, CD9 expression, and morphological distribution of gastric fluid EVs between gastric cancer patients and healthy individuals. This study has important scientific significance and clinical value.

1.It is necessary to clarify that the sample size is small (especially the healthy control group with n=25). It is suggested to increase the sample size.

2.The authors only used cryo-electron microscopy (cryo-EM) to analyze extracellular vesicles (EVs) from gastric juice sources, without verification and comparison with other imaging techniques. It is suggested that the authors consider using other advanced imaging techniques to verify the results of cryo-EM and to enrich the characterization of EVs.

3.The paper discusses the distribution characteristics of CD9-positive and CD9-negative EVs but does not fully explain the biological significance of these findings. The authors are advised to deepen this area.

4.What functions may be associated with different EV subtypes? What are the associations with gastric cancer progression or clinical diagnosis? The potential mechanisms linking CD9 expression to EV size should be explained in detail.

5.The paper should make more specific clinical translational recommendations, such as the development of EV diagnostic detection methods or the identification of novel biomarkers, which can be further discussed in the conclusion section.

6.Regarding the discussion on clinical translation, the analysis of clinical challenges such as immune clearance and large-scale production is relatively shallow. It is suggested to add content discussing the industrial challenges of liposomes, such as stability and quality control.

7.Chart optimization: The significance mark of Figure 2C should be clearer; the Tukey method in Figure 5E is suggested to supplement the specific p-value of inter-group comparison.

8.Some of the references are older and need to be optimized by the authors.

Author Response

Dear Reviewer,

We sincerely thank you for the careful evaluation of our manuscript and for the thoughtful comments. We have addressed your points and hope that the revisions will significantly improve the quality of the work. We provide a point-by-point response in the attached file.

Reviewer 2 Report

Comments and Suggestions for Authors

1. The study identifies six morphological EV subtypes, including multilayered vesicles and vesicle sacs. Could the authors provide metrics for inter-rater reliability or reproducibility in the manual classification process? Given the subjective nature of morphology-based assessment, such metrics would enhance the robustness of the findings.

2. Were the atypical EV morphologies—such as multilayered vesicles or vesicle sacs—quantitatively assessed across gastric cancer and healthy donor samples? If not, could the authors clarify whether these structures appear enriched in either group, and whether any trends were observed?

3. The results indicate that individual EVs are larger than nested EVs of the same subtype. Could the authors discuss potential biological mechanisms underlying this size discrepancy? Additionally, might this observation be influenced by the EV isolation method used?

4. In Figure 5E, CD9-positive EVs are significantly larger than CD9-negative EVs across all groups. Were similar size trends observed within specific morphological subtypes or among the six EV categories? Clarifying this could help determine whether size differences are independently associated with morphology or CD9 expression.

5. Given that cryo-EM identified a broader range of EV sizes—particularly <70 nm and >350 nm—than NTA, how do the authors account for potential discrepancies in size-dependent findings (e.g., CD9-positive vs. CD9-negative EVs) derived solely from NTA data? Could limitations in NTA sensitivity introduce bias in the reported size distributions?

6. Consider more clearly articulating the diagnostic advantages of gastric juice-derived EVs over blood-based EVs early in the introduction. Emphasizing their specificity to the tumor microenvironment and distinct molecular cargo could better frame the study's clinical relevance.

Author Response

(The authors gave the same response as above.)

Round 2

Reviewer 1 Report

Comments and Suggestions for Authors

This manuscript can be accepted for publication in present form.

Reviewer 2 Report

Comments and Suggestions for Authors

Accept this article in its current format.